# Throw out the Map: Neuropathogenesis of the Globally Expanding California Serogroup of Orthobunyaviruses

**DOI:** 10.3390/v11090794

**Published:** 2019-08-29

**Authors:** Alyssa B. Evans, Karin E. Peterson

**Affiliations:** Laboratory of Persistent Viral Diseases, Rocky Mountain Laboratories, National Institute of Allergy and Infectious Diseases, National Institutes of Health, Hamilton, MT 59840, USA

**Keywords:** orthobunyavirus, neuroinvasion, neurovirulence, pathogenesis, arbovirus

## Abstract

The California serogroup (CSG) comprises 18 serologically and genetically related mosquito-borne orthobunyaviruses. Of these viruses, at least seven have been shown to cause neurological disease in humans, including the leading cause of pediatric arboviral encephalitis in the USA, La Crosse virus. Despite the disease burden from these viruses, much is still unknown about the CSG viruses. This review summarizes our current knowledge of the CSG viruses, including human disease and the mechanisms of neuropathogenesis.

## 1. Introduction

The California serogroup (CSG) is a closely related group of orthobunyaviruses in the family *Peribunyaviridae* of the order *Bunyavirales*. Like all bunyaviruses, the CSG viruses contain tri-segmented, negative sense RNA genomes containing Large (L), Medium (M), and Small (S) genome segments. Classification of CSG viruses has fluctuated over time, as better serological assays and genetic techniques developed. California encephalitis virus (CEV) is the prototype member and was first isolated from mosquitoes in California in 1943 [1]. Currently, there are 18 recognized members of the CSG based on serological and genetic evidence. In addition to CEV, these are La Crosse virus (LACV), Snowshoe hare virus (SSHV), Tahyna virus (TAHV), Jamestown Canyon virus (JCV), Inkoo virus (INKV), Chatanga virus (CHATV), Keystone virus (KEYV), Jerry Slough virus (JSV), Lumbo virus (LUMV), Melao virus (MELV), San Angelo virus (SAV), Serra do Navio virus (SDNV), South River virus (SORV), Trivittatus virus (TVTV), Morro Bay virus (MBV), Achiote virus (ACHOV), and Infirmatus virus (INFV). Guaroa virus was briefly categorized as a CSG virus, however follow-up studies have since indicated it is a distinct virus [2]. 

All members of the CSG are presumed arboviruses, as all have been isolated from mosquitoes [3,4,5,6,7,8,9,10,11,12,13,14,15]. However, their vector, host ranges, geographic distribution, and ability to cause disease in humans differ, including their ability to cause neuroinvasive disease. This review will summarize our current knowledge of the members of the CSG and their ability to cause neuroinvasive disease in humans.

## 2. Relationships of the CSG Viruses

The CSG viruses are antigenically and genetically related, however determining their exact relationships with each other has been challenging. Based on traditional serological tests, three complexes within the CSG have been described: 1) the California encephalitis complex containing CEV, LACV, SSHV, CHATV, SAV, TAHV, LUMV, MBV, and INKV; 2) the Melao complex containing MELV, KEYV, SDNV, SORV, JCV, and JSV; and 3) the Trivittatus complex, containing TVTV and the recent additions of ACHOV and INFV [8,14,16,17]. Genetic characterization of the CSG viruses by phylogenetic analysis has shown that genetic relationships vary by genomic segment, viruses, strains, and methodology used in analyses. In recent years, several CSG phylogenies have been constructed using a variety of methods and virus strains, with some topology discrepancies [8,14,18,19]. However, some general conclusions of genetic relatedness can be made from these studies. Overall, the CSG viruses separate phylogenetically into three distinct groups, similar to but not identical to the serocomplexes: 1) CEV group containing CEV, LACV, SSHV, CHATV, SAV, TAHV, LUMV, and MBV; 2) MELV group containing MELV, KEYV, SDNV, SORV, JCV, JSV, and INKV; 3) TVTV group containing TVTV, ACHOV, and INFV [8,14,18,19]. MBV has only been included in phylogenies based on the S segment and appears to group most closely with CEV, which is how we have classified it in this review [17,19]. However, in one topology, MBV was distinct from the CEV clade [14]. These studies show that antigenic relatedness does not necessarily correlate with genetic relatedness, as INKV is classified in the CEV by serology, but the MELV group by phylogenetic analysis [14,18,20].

Based on results from the phylogenic studies of the CSG viruses in [8,14,18,19], within the CEV clade, SSHV, CHATV, and LACV are most closely related to each other, and LUMV and TAHV are closest to each other. The relationship of CEV and SAV within the CEV clade varies by phylogeny. In the MELV clade, JCV, JSV, and INKV are the most closely related to each other, and these three also form a clade with SORV. The relationship of MELV, SDNV, and KEYV is more variable within the clade. In all phylogenies, TVTV forms a distinct clade, and in the study first describing ACHOV and INFV as members of the CSG, these two viruses always grouped most closely with TVTV, expanding the group to three members [8,14,18,19]. These relationships are visually compiled and summarized in Figure 1. 

## 3. Geographic Distribution

The CSG viruses expand across the globe, however the geographic distribution of each CSG virus differs, likely restricted by the range of their vertebrate hosts and mosquito vectors. Figure 2 summarizes the reported geographic distribution of each CSG virus based on human cases, seroprevalence studies, and virus isolations from mosquitoes. Eleven CSG viruses were detected in the USA, including LACV, SSHV, JCV, KEYV, TVTV, CEV, SORV, MBV, JSV, SAV, and INFV (Figure 2; [3,4,5,9,14,15,22,23,24,25,26,27,28,29,30,31,32,33,34,35,36,37,38,39,40,41,42,43,44,45,46,47,48]. SSHV, JCV, and TVTV have also been detected in Canada, and SORV has also been found in the Yucatan Peninsula of Mexico (Figure 2; [23,49,50]). ACHOV was isolated from mosquitoes in Panama [14], MELV in Trinidad and Brazil [13,51], and SDNV in Brazil [52]. INKV and CHATV are primarily found in Scandinavia and Russia [53,54,55,56], while TAHV is found throughout Europe, Russia, Asia, and Africa [53,57,58,59,60,61,62]. LUMV is the only CSG member found, thus far, exclusively in Africa [12,61]. Some of the CSG viruses have only been isolated a limited number of times, so it is unclear if all 18 viruses are actively circulating. The exact range of the CSG viruses is unclear, as limited studies have been conducted and they are not consistently monitored, so the range of these viruses may be more widespread than reported. Furthermore, as climate change and globalization alters the territories of mosquitoes, the geographic distribution of the CSG viruses is likely to change as well. 

## 4. Vector and Host Range

The CSG viruses have been isolated from a variety of mosquito species, primarily in the *Aedes* genera. Mosquito taxonomy has been a controversial topic in the 2000s, where many of the *Aedes* species were reclassified, but for the purpose of this review we will use the mosquito classification proposed by Wilkerson et al. in 2015 and used in Walter Reed’s Systematic Catalog of *Culicidae* (mosquitocatalog.org) [63]. Many CSG viruses have been isolated from a variety of different species and genera, however, not all mosquitoes that carry a virus are competent vectors. It can be difficult to establish a primary vector for arboviruses, and primary vector species can vary by region even for the same virus. However, for some of the CSG viruses the primary vector is well established. The CSG viruses and their potential vector relationships are summarized in Figure 1. Primary vectors for several of the CSG viruses have been established based on field isolations and transmission studies using suckling mice and other mammals. These include *Ae. triseriatus* for LACV, *Ae. atlanticus* for KEYV, *Ae. trivittatus* for TVTV, *Ae. squamiger* for MBV, and *Ae. dorsalis* for CEV (Figure 1), [5,10,11,26,30,42,64,65,66,67,68,69,70,71]). *Ae. canadensis* and *Ae. albopictus* have been proposed as additional vectors of LACV [72]*,* [73]. JCV and SSHV have both been isolated from a wide variety of mosquitoes (and horseflies in the case of JCV), and several mosquito species are capable of transmitting these viruses in laboratory experiments (Figure 1) [5,26,30,74,75,76,77,78,79,80,81,82]. It is unclear if JCV and SSHV utilize multiple primary vectors, or if they are simply easily transmittable to suckling mice under laboratory conditions. INKV has nearly exclusively been isolated from *Ae. communis*, thus it has been assumed to be INKV’s primary vector [7,54,83]. TAHV has been isolated from *Aedes, Culiseta, Culex*, and *Anopheles* species, and *Ae. vexans* and *Ae. cantans* have been implicated as primary vectors [53,84,85]. 

Several of the CSG viruses have only been isolated once or a handful of times. ACHOV was isolated from a pool of *Aedes* species, INFV from *Ae. infirmatus*, JSV from *Cs. inornata*, SDNV from *Ae. fulvus*, and LUMV from *Ae. pembaensis*. SAV, SORV, and MELV have been isolated from several mosquito species; SAV from *An. pseudopunctipennis*, *Ps. columbiae*, *Ae. atlanticus*, and *Ae. infirmatus*, SORV from *Anopheles, Culex, Aedes*, and *Culiseta* species, and MELV from *Ae. scapularis*, *Ae. serratus*, *Ps. ferox*, and *Culex* species [5,11,13,48,50,86,87]. CHATV was isolated from pools of unidentified mosquitoes, and positive pools were sequenced for mosquito mitochondrial cytochrome c oxidase to try to identify mosquitoes in the pool. From the pooled sequencing, *Ae. cantans, Ae. annulipes*, *Ae. cinereus*, *Ae. communis, Ae. punctor* were suggested as possible vectors [8]. Because of the limited isolations of these viruses, their primary vectors are still unknown.

Eight of the CSG viruses, LACV, CEV, JCV, TAHV, KEYV, TVTV, SSHV, and MBV are transovarially transmitted in their mosquito vectors, which likely contributes to their survival over winter ([30,67,69,88,89,90,91,92,93,94,95]. SAV has also been shown to be transmitted transovarially in *Ae. albopictus*, though the virus has not been isolated from this mosquito in nature, so the significance of transovarial transmission in SAV’s natural life cycle remains unclear [96]. Additionally, INKV RNA was detected in *Ae. communis* larvae, suggesting transovarial transmission of this virus as well [97]. Too little information is available on the other CSG viruses to know if they are transovarially transmitted, however it is possible that transovarial transmission is a common feature among the CSG viruses. 

The CSG viruses have been shown to utilize a variety of mammal hosts in their natural life cycle, and suspected host and reservoir species are summarized in Figure 1. Based on serological surveys and the development of viremia after experimental inoculations, small mammals such as squirrels, chipmunks, other rodents, and/or hares have been implicated as hosts for LACV, CEV, SSHV, TAHV, KEYV, and possibly JSV (Figure 1, [1,11,53,65,67,92,98,99,100]). In addition to experimental viremia, SSHV has been isolated from hares in the wild during winter months, suggesting long-term infection of these animals and potentially a second over-wintering strategy for SSHV [11]. Experimental infections with LUMV resulted in viremia in vervet monkeys, but information on natural infections is lacking [12]. JCV, thus far, appears to be the only CSG virus to primarily utilize large mammals as its reservoir. White-tailed deer are its primary host, but other large ungulates such as moose, elk, and bison have also been implicated as amplifying hosts and neutralizing antibody has been detected in domestic animals such as sheep, cattle, and horses [30,38,101,102,103]. Seropositivity to SSHV has been demonstrated in large mammals as well, however, experimental infection of deer with SSHV did not produce viremia [99,101,103,104,105].

For INKV, SAV, and TVTV, vertebrate hosts have been speculated on based solely on serological surveys, however the role of these seropositive animals as reservoirs is unclear. For INKV, high rates of seropositivity have been shown in reindeer, moose, cows, red foxes, and hares; neutralizing antibody (NAb) to SAV has been reported in coyotes, raccoons, and opossums; and NAb to TVTV has been found in rabbits, fox squirrels, opossums, and raccoons [7,11,41]. There is little to no evidence as to the vertebrate hosts of MELV, CHATV, SORV, SDNV, MBV, INFV, and ACHOV.

## 5. Replication

Few studies have been done specifically looking at the replication cycle of CSG viruses. What is known about the CSG virus genomic organization and replication cycle is based primarily on studies of LACV and SSHV. Additional information about the CSG replication cycle has been inferred from studies of Bunyamwera virus (BUNV), the *Orthobunyavirus* prototype, and other orthobunyaviruses (reviewed in [106]). The CSG viruses, like all orthobunyaviruses, contain three single-stranded negative sense RNA genome segments. The exact length of each genome segment varies by virus, but typically are about 6.9 kb (L), 4.5 kb (M), and 1.0 kb (S) [106]. Each genome segment is flanked by complementary 5’ and 3’ UTRs, which circularize each genome segment by forming a panhandle structure that is used as the viral promoter [107,108]. Encapsidation of the genome segments is initiated by interactions between the 5’ end of the genomic segments and the Nucleocapsid (N) protein, resulting in ribonucleoprotein (RNP) complexes [109,110,111,112]. These genomic RNPs associate with the L protein in the virion through interactions with the UTRs [106]. The L protein is encoded by the L segment; the M segment encodes the two envelope glycoproteins (Gn [previously G2] and Gc [previously G1]) and a non-structural protein (NSm), and the S segment encodes the N protein and a non-structural protein (NSs) [113,114,115,116]. M encodes a single open-reading frame (ORF) resulting in the glycoprotein precursor (GPC) polyprotein in the order Gn-NSm-Gc [117]. In studies of BUNV, it was shown that GPC is processed into mature proteins by internal signal peptides and host signal peptide peptidase [118]. N and NSs are encoded by overlapping reading frames on the S segment, with NSs translation resulting from leaky ribosomal scanning [119,120]. 

Much of the process of CSG virus entry into cells is still unknown. The CSG viruses enter cells via receptor-mediated endocytosis initiated by binding of the Gn/Gc heterodimer, likely mediated by Gc, with a currently unknown cell receptor [106,121,122]. DC-SIGN has been implicated as a potential LACV receptor [123]. However, DC-SIGN is likely not the only, or primary, receptor as its expression is limited to certain Dendritic cell and macrophage types and LACV, in the CNS, primarily infects neurons [124,125]. Studies of the related Akabane and Schmallenberg (SBV) orthobunyaviruses have shown that heparan sulfate facilitates viral attachment to cells, however further studies are needed to know if this is an important attachment factor for the CSG viruses as well [126]. 

Studies of LACV have shown that upon attachment, the virus enters the cell via clathrin-mediated endocytosis [127]. Uncoating of LACV occurs in the early endosome where the change in pH induces conformational changes of the envelope proteins mediated by the Gc fusion peptide, resulting in the release of the RNPs into the cytoplasm [127,128,129,130,131]. LACV transcription is primed via cap-snatching of the 5’ 7-methylguanylate capped ends of host-derived mRNAs, in a process facilitated by an endonuclease domain on the L protein [132,133]. Transcription of LACV, like all Bunyaviruses, requires concurrent host translation to prevent premature termination by the RNA-dependent RNA polymerase (RdRp), and results in viral mRNAs lacking a poly(A) tail [106,134,135,136]. Studies of BUNV and Oropouche (OROV) orthobunyavirus have shown that these viruses form Golgi-derived viral factories for replication [137,138]. Genome replication is primer-independent and is carried out by the RdRp, encoded by the L protein, on full-length anti-sense RNPs [106].

Orthobunyavirus virions assemble, mature, and bud at the membrane of the Golgi [106]. For BUNV, the viral RNPs are trafficked to the glycoprotein-expressing Golgi membrane, and genome packaging is facilitated by the UTRs [106,109]. For OROV, remodeling of the Golgi membranes to create viral factories requires endosomal complex required for transport (ESCRT) machinery, and this process is likely required for virus assembly and budding [138]. Budding is facilitated by the interactions of the RNPs with Gn and Gc, virions are released from the cell via exocytosis, and the glycoproteins undergo a final maturation step upon exit [139,140,141,142]. The CSG viruses likely utilize a similar mechanism, as Gn and Gc of LACV are processed in the Golgi [121,143]. However, further studies specifically on the CSG viruses are needed to elucidate differences in replication within and between the CSG viruses and other orthobunyaviruses.

## 6. Human Infection and Disease 

Several CSG viruses have been shown to cause disease in humans, ranging from mild rashes to severe neurological disease (Figure 1). However, infection and disease rates are difficult to determine as serosurveys have primarily focused on the CSG viruses capable of causing human disease and are complicated by the cross-reactivity among CSG viruses. Moderate to high seroprevalence rates in endemic regions of many CSG viruses suggest that these viruses frequently infect humans, but that symptomatic disease is rare. Depending on the region sampled, reported seroprevalence rates for CSG members are up to 27% for LACV, JCV, CEV, KEYV, LUMV and TVTV, up to 42% for SSHV, 24–50% for INKV and/or CHATV, and up to 80% for TAHV [1,8,12,25,47,53,54,144,145,146,147,148,149,150], Even when clinical cases do arise, it can be difficult to diagnose the specific CSG virus responsible due to the high amount of antibody cross-reactivity within the serogroup. Additionally, historical cases of CSG viral disease may have been misattributed to the wrong CSG virus due to less sensitive diagnostic techniques, lack of inclusion of relevant CSG virus comparisons, or they were performed prior to the identification of other CSG viruses. 

Of the 18 CSG viruses, eight have caused confirmed human disease (Figure 1). CEV, LACV, SSHV, JCV, TAHV, INKV, and CHATV have been shown to cause neuroinvasive disease [1,55,84,151,152,153]. KEYV was associated with one case of febrile illness and diffuse rash without CNS involvement [146]. TVTV was retrospectively implicated in one case of human neurological disease in New York in 1981, but no additional cases have been reported since, so it is difficult to say if TVTV causes human disease [43]. Among the CSG viruses that cause neurological disease, the incidence and ages of susceptibility vary. 

In North America, LACV, JCV, and SSHV are the main disease-causing CSG viruses. LACV traditionally has caused the most cases of neuroinvasive disease annually, primarily in children, and is the leading cause of pediatric arboviral encephalitis in the USA with an average of ~70 reported neuroinvasive cases per year [154]. In recent years, JCV has been responsible for an increasing number of neuroinvasive cases, which may be due to better arboviral reporting and testing, or it may reflect an emergence of this pathogen. In the USA, the CDC started routine JCV IgM testing in 2013 and since then ~25 neuroinvasive cases are reported in the US annually [22]. JCV neuroinvasive cases also occur in Canada every year [23,155]. Unlike LACV, JCV appears to affect more adults than children [24,43]. SSHV causes several reported cases of neuroinvasive disease annually in Canada, primarily in children [23,152]. Since the first reports of CEV neuroinvasive disease in three people in 1945, only one other confirmed case of CEV has been reported, a 65-year-old man with blurred vision and dizziness in 1996 [1,156]. All four CEV cases occurred in California. 

In Europe, INKV, CHATV, and TAHV cause human disease. TAHV disease primarily manifests as a flu-like illness called “Valtice fever”, and only rarely causes neuroinvasive disease [53,157,158]. Like LACV and SSHV, TAHV primarily causes disease in children [53]. Neuroinvasive cases of INKV and CHATV were described in Finland during a retrospective study of serum samples from patients with suspected viral neuroinvasive disease [55]. INKV caused neuroinvasive disease in both children and adults, but children appeared to have more severe disease than adults [55]. There were three suspected and confirmed CHATV cases, all of which were adults (≥40 years old) [55]. 

CSG virus neuroinvasive disease symptoms include headache, fever, dizziness, vomiting, muscle weakness, stiff neck, seizures, encephalitis, meningoencephalitis, and meningitis [23,55,59,154,159,160]. While most patients with CSG virus neuroinvasive disease fully recover, some cases result in long-term and serious sequelae including seizures, hemiparesis, cognitive deficits, and behavior changes [154,161,162]. LACV has a reported case fatality rate of up to 1.9% in children [35]. Fatalities from the other CSG viruses have not been reported. Because mild illnesses are not included and testing for CSG viruses is not routinely done unless arboviral disease is suspected, disease from the CSG viruses is likely underreported. 

## 7. Neuropathogenesis – Experimental Infections of Animals

Despite the ability of some of the CSG viruses to cause neuroinvasive disease, and new cases occurring every year, our understanding of the pathogenesis of these viruses is still limited. Traditionally, CSG virus pathogenesis was evaluated using a suckling mouse model via intracranial (IC) inoculation or mosquito transmission. KEYV, LUMV, MELV, TVTV, JCV, SSHV, CEV, TAHV, and LACV have all been shown to cause disease and death in suckling mice [3,12,13,30,51,67,68,78,128,163]. Suckling mice were particularly useful for mosquito transmission studies, but because they are so susceptible to CSG virus infection; they are not a very informative model for pathogenesis studies. 

Weanling (3–4-week-old) and adult mice are also susceptible to many CSG viruses; however, susceptibility depends on inoculation route and dose. Pathogenicity of viruses that cause neurological disease can be evaluated by a number of parameters. Neuroinvasion describes the ability of the virus to enter the brain from the periphery, regardless of the development of disease. Neuroinvasive disease is evaluated by the ability of the virus to enter the CNS from the periphery and cause disease. Neurovirulence describes the ability of the virus to replicate in the brain following a direct route of inoculation, such as IC or intranasal (IN), and neuropathogenicity is the ability of the virus to cause neurological disease. 

Of the CSG viruses that have not been shown to cause human neurological disease, all that have been tested in weanling mice are neurovirulent, however not all are neuroinvasive. Following IC inoculation of weanling mice, LUMV, MELV and KEYV all cause disease indicating neurovirulence [9,12,13]. However, LUMV is not pathogenic following IP inoculation indicating it is not neuroinvasive. Some strains of MELV and SAV can cause disease in weanling mice or hamsters following IP inoculation, indicating these viruses are neuroinvasive and neurovirulent [12,51,164]. 

Interestingly, similar patterns of pathogenicity are observed in weanling mice inoculated with the CSG viruses known to cause human disease. Weanling mice are susceptible to disease from CEV, JCV, and INKV following IC or IN inoculation, but not following IP inoculation, indicating these viruses are neurovirulent, but not neuroinvasive, in mice [3,21,165]. LACV, SSHV, and TAHV all cause neuroinvasive disease in weanling mice following IP inoculation. Of note, the ability of TAHV to cause neuroinvasive disease varies by strain and requires a high dose, suggesting that it is less virulent than LACV or SSHV [21,125,166,167,168]. Neuroinvasive disease caused by LACV, SSHV, and TAHV is age-dependent, as adult and aged mice are resistant to neuroinvasive disease after IP inoculation [21,163,167]. However, LACV, SSHV, TAHV, and JCV are highly neurovirulent and cause disease in adult and aged mice when inoculated IN. In contrast, INKV caused disease in only about 20-25% of mice when inoculated at a high dose IN in adult mice, suggesting this virus differs in neuropathogenesis compared to the other CSG viruses that cause human disease [21]. 

The results of the mouse infection studies reflect, to an extent, human disease caused by the CSG viruses. LACV, SSHV, TAHV, and INKV all have been reported to cause more serious neuroinvasive disease in children than adults. LACV causes the most reported neuroinvasive cases annually, followed by SSHV, then TAHV and INKV. At a high dose (10^5^ PFU virus), LACV, SSHV, and TAHV all cause neuroinvasive disease in the majority of weanling mice, however LACV and SSHV, but not TAHV, maintain their pathogenicity at a low dose of 10^3^ PFU [21,167]. Additionally, LACV causes disease in nearly 100% of mice inoculated but SSHV causes disease in ~60–80% of mice [21]. These results may reflect differences in virulence and account for the higher number of LACV neuroinvasive cases reported in humans compared to SSHV and TAHV. In humans, INKV causes few neuroinvasive cases and disease is more severe in children than adults. In mice, INKV did not cause neuroinvasive disease, however, weanling mice had a higher incidence of disease than adult mice when inoculated IN, correlating with reported disease in humans [21]. Because some of the CSG viruses that have not been reported to cause neurological disease in humans can cause disease when inoculated into animals IC, it is possible that neurotropism is a shared trait among the CSG viruses. The ability to enter the brain may determine which CSG viruses cause neuroinvasive disease in humans. It is also possible that some cases of neurological disease in humans have been misattributed to the wrong CSG virus due to the high level of cross-reactivity among the group. The development of more specific diagnostic tests will provide a more accurate account of which CSG viruses cause neurological disease in humans.

In experimental infections of rhesus monkeys inoculated subcutaneously or intramuscularly with LACV, TAHV, or JCV, none of the monkeys developed disease [125,165,168]. However, nearly all monkeys developed high neutralizing antibody titers, indicating they were productively infected [125,165,168]. These results suggest that CSG virus infection in monkeys may be akin to human infections, where infection and NAb responses happen readily, but neuroinvasive disease is a rare event.

## 8. Neuropathogenesis – Molecular Mechanisms

Much of what is known mechanistically about CSG virus pathogenesis is based on studies of LACV and has focused on the host interferon (IFN) response. The IFN response, primarily type-I IFNs α and β, has been shown to be important in many viral infections, including LACV [169]. IFNs are activated through a variety of pathways in response to RNA viruses. These include the RIG-I-like receptor (RLR) and Toll-like receptor (TLR) pathways. In general, RLR pathways are activated in response to RNA viruses in the cytoplasm, and signal through mitochondrial antiviral-signaling proteins (MAVS, aka IPS-1) and interferon regulatory factor (IRF)3 and IRF7 [170,171,172]. TLRs 3, 7, and 9 are endosomal and can recognize viral RNA within endosomes and use the myeloid differentiation factor-88 (MyD88) adaptor protein in the case of TLR7 and TLR9 [173]. Signaling through these pathways stimulates the production of Type-I IFNs which are released from cells, bind to cellular IFN receptors (IFNAR), then signal through the JAK–STAT (Janus kinase-signal transducer and activator of transcription proteins) pathway to induce the expression of IFN-stimulated genes and effector molecules with antiviral properties [173]. 

Investigations into the mechanism of age-dependent susceptibility to LACV in mice have shown that the IFN response has an important role. While wild-type adult mice are resistant to neuroinvasive disease, adult *Ifnar-/-* mice are highly susceptible to neuroinvasive disease following IP inoculation of LACV, indicating that the type I interferon (IFN) response plays a crucial role in protection [169,174,175]. Adult *Irf3/Irf7-/-, Mavs-/-*, and *Unc93b1*3D (deficient in TLR3/7/9) mice, but not *Myd88-/-* mice, have an increased susceptibility to neuroinvasive LACV disease, indicating that protection against LACV requires both RLRs and TLRs [167]. In wild-type (WT) mice, a robust peripheral IFNα4 and IFNβ response was observed in adult mice, but not in weanlings, and myeloid DCs were responsible for their expression. In weanling mice, mDCs had little production of type-I IFNS, leading to the susceptibility of young mice to neuroinvasive disease [167]. The IFN effector protein myxovirus resistance protein 1 (MxA) has been shown to be capable of restricting LACV growth in vitro, and MxA transgenic *Ifnar-/-* mice are less susceptible to neuroinvasive disease, indicating an important role for MxA in protection [176,177]. Additionally, the adaptive immune response also plays a role in age-dependent susceptibility to LACV neuroinvasive disease. In adult mice, depletion of CD4+ helper and CD8+ cytotoxic T cells or depletion of B cells results in increased susceptibility of mice to neurological disease, indicating a protective role of lymphocytes in adult mice [178]. In contrast, when T and B cells are depleted in weanling mice, neuropathogenesis is not affected despite infiltration of these cells into the brain during disease in WT mice [178]. 

In addition to mediating age-dependent susceptibility to CSG viruses, immune responses may also influence the susceptibility of host species. In studies comparing restriction of LACV, which infects humans but not ruminants, and SBV, which infects ruminants but not humans, by human and sheep restriction factor BST-2/tetherin, LACV was restricted by sheep BST-2, but not human BST-2, and the reverse was true for SBV, suggesting a mechanism of orthobunyavirus host determination [179]. However, whether BST-2 restriction plays a role in age-related CSG virus pathogenesis, or if restriction differs between CSG viruses, has not been investigated.

Dissemination of the CSG viruses from the periphery to the brain likely occurs with the virus first replicating near the site of entry (via mosquito bite or inoculation) in the muscle, lymph nodes, or spleen [125,163,180,181]. Infectious LACV and RNA can be found at low levels in many tissues throughout the body after IP inoculation of weanling and suckling mice, suggesting either a wide tissue tropism for the virus, or representing virus in the blood within these tissues [125,163]. In support of the latter, short viremia is observed in weanling mice, likely contributing to its dissemination and presence in multiple tissues, but viral antigen staining has not been reported for peripheral tissues, suggesting a lack of, or limited, replication in peripheral tissues ([125], unpublished observations). Entry of LACV into the CNS is mediated by blood-brain-barrier (BBB) breakdown via vascular leakage of the capillaries in the olfactory bulb, which allows LACV entry into the brain via a hematogenous route rather than axonal transport in olfactory sensory neurons [182]. Mice deficient in the TAM receptor Mertk, which is involved in innate immune regulation, are more susceptible to LACV neuroinvasive disease [183]. This susceptibility was associated with increased BBB permeability, suggesting TAM receptors may be involved in BBB integrity during LACV infection [183]. 

Once in the CNS, the primary site of replication for the CSG viruses is neurons, and the CSG viruses readily replicate in neuronal cultures in vitro [21,125,163,166]. Co-cultures of neurons and astrocytes have shown that astrocytes are susceptible to LACV infection in vitro, however this does not appear to be a main target cell in vivo [184]. LACV, SSHV, TAHV, and JCV all replicate extensively throughout the mouse brain, however INKV’s replication is limited [21]. Neurological disease from LACV is the result of neuronal damage and cell death, and this damage is mediated by sterile alpha and TIR-containing motif 1 (SARM1), a MyD88-related protein expressed at high levels in neurons and associated with multiple forms of neuronal death [185], [186]. In vitro, siRNA to *Sarm1* reduced LACV-induced death in neuronal cultures, and in vivo, weanling *Sarm1-/-* mice were less susceptible to neuroinvasive disease following LACV inoculation, despite similar amounts of virus in the brains of WT and *Sarm1-/-* mice [186]. In neuronal cultures in vitro, SSHV, TAHV, JCV, and INKV also induce cell death, however, depending on cell type, not typically to the same degree as LACV [21]. Further studies are needed to determine if neuronal cell death caused by the other CSG viruses is similarly mediated by SARM1. 

In weanling mice inoculated with LACV, infiltrating monocytes/macrophages are associated with areas of infection [125]. Characterization of these cells demonstrated that the majority were inflammatory monocytes, which are defined by their high expression of the cell surface marker Ly6C, as well as CCR2, a chemokine receptor that is involved in monocyte trafficking from the bone marrow to the blood. Surprisingly, unlike other neuroinvasive viral infections, CCR2 is not required for monocyte recruitment to the blood following LACV infection [187]. Within the brain, however, iMO trafficking to sites of infection is CCR2-dependent [187]. Interestingly, CSG viruses vary in their iMO recruitment mechanism. JCV induces a similar CCR2-independent monocyte recruitment mechanism from the bone marrow to the blood [187]. However, iMO recruitment from the bone marrow during TAHV infection is CCR2-dependent [187]. These results are particularly interesting in the respect that LACV and TAHV are more closely related to each other than to JCV; LACV and TAHV are both neuroinvasive in weanling mice and JCV is not, yet LACV and JCV share a monocyte recruitment mechanism distinct from TAHV. This highlights the need for studies to better understand the mechanistic differences of pathogenesis between the CSG viruses. 

Studies investigating the virulence factors of the CSG viruses have shown that the M segment, L protein, and NSs all contribute to LACV virulence. In experiments that utilized reassortants between a virulent LACV strain and a lowly virulent TAHV strain (as determined by neuroinvasion in suckling mice), neuroinvasiveness primarily mapped to the M segment [188]. However, some intermediate phenotypes were observed suggesting compensatory mutations could exist in the L and/or S segments [188]. In additional studies using an attenuated CSG virus clone, neurovirulence was primarily determined by the L segment, but again the neurovirulence phenotype also appeared dependent on the other genomic segments [180,189]. More recently, a recombinant virus containing the LACV backbone with the M ORF from JCV (rLACV/JCV) showed that the chimeric virus was highly attenuated and unable to cause disease after either IP or IC inoculation at high doses, a surprising result considering both parental LACV and JCV viruses were highly neurovirulent [190]. Taken together, the results of these studies suggest that interactions between multiple CSG viral proteins encoded from multiple genomic segments determine CSG virus virulence. Individual CSG virus proteins may have unexpected phenotypes when expressed outside of their specific genetic context. This may be an advantageous feature in terms of vaccine development, as the highly attenuated rLACV/JCV virus was highly immunogenic and when mice and monkeys were immunized with it, they were protected from subsequent LACV, TAHV, and JCV challenge [190]. However, if reassortants between the CSG viruses can have unique phenotypes from either parental virus, it is possible that more virulent novel reassortants may arise. 

An important virulence factor for LACV is NSs. Utilizing a recombinant LACV virus with an NSs deletion, it was shown that NSs antagonizes the type-I IFN response in vitro and in vivo [191,192]. The NSs deletion mutant strongly induced type-I IFN responses in vitro and in vivo, whereas WT LACV did not induce high levels of type-I IFN [192]. NSs appears to not play a role in pathogenesis outside of IFN antagonism, as no difference in virulence was observed between WT and NSs deletion LACV viruses in cells and mice deficient in type-I IFN responses [192]. NSs antagonizes the IFN response by targeting and degrading host RNA Polymerase II [192,193]. NSs did not affect LACV’s ability to infect neurons in the mouse CNS, and ultimately most mice still succumbed to disease from the NSs deletion virus [192]. The authors speculated that this may be due to a lack of an appropriate IFN effector protein, as the MxA homologue in mice, Mx2, is non-functional in C57BL/6 mice, the strain used for the studies [192]. It remains to be seen whether differences in virulence between the CSG viruses may be due to differences in the ability of their NSs to antagonize host IFN responses. 

The majority of pathogenesis studies of the CSG viruses have focused on LACV. Studies that have examined multiple CSG viruses have often found differences between CSG viruses. Further studies are clearly needed to elucidate the mechanisms of pathogenesis for the different neuroinvasive CSG viruses. 

## 9. Conclusions and Future Perspectives

The California serogroup of orthobunyaviruses comprises a large, related group of mosquito-borne viruses with differing geographical distributions, vector and host ranges, and abilities to cause human disease. Despite the continual, and possibly increasing, disease burden from the CSG viruses, much is still unknown. Of particular interest is why only a subset of infections in humans result in neurological disease, and why susceptibility to specific viruses differs with age. Research efforts have primarily focused on LACV and a handful of other CSG viruses. From these studies, we know that the mechanism of LACV pathogenesis can vary from other CSG viruses, highlighting the importance of future research into LACV and other CSG viruses. Additionally, as climate change continues, the geographical range and abundance of arboviral mosquito vectors is also changing [194]. Many of the human disease-causing CSG viruses already exist in overlapping geographical regions and share some vector and host species. As mosquito ranges expand, the potential for coinfections and novel reassortant CSG viruses increases. Laboratory infections of mosquitoes with LACV and SSHV have shown that it is possible to recover reassortant viruses from coinfected mosquitoes, suggesting the possibility of the occurrence of highly virulent novel reassortants in nature [195,196]. Elucidating the differences between the neuroinvasive CSG viruses and the non-neuroinvasive CSG viruses, and differences in virulence between the neuroinvasive CSG viruses, will help us understand how these viruses cause disease. A better understanding of the underlying mechanisms of pathogenesis of different CSG viruses will help in the development of treatments and vaccines.

## Figures and Tables

**Figure 1 viruses-11-00794-f001:**
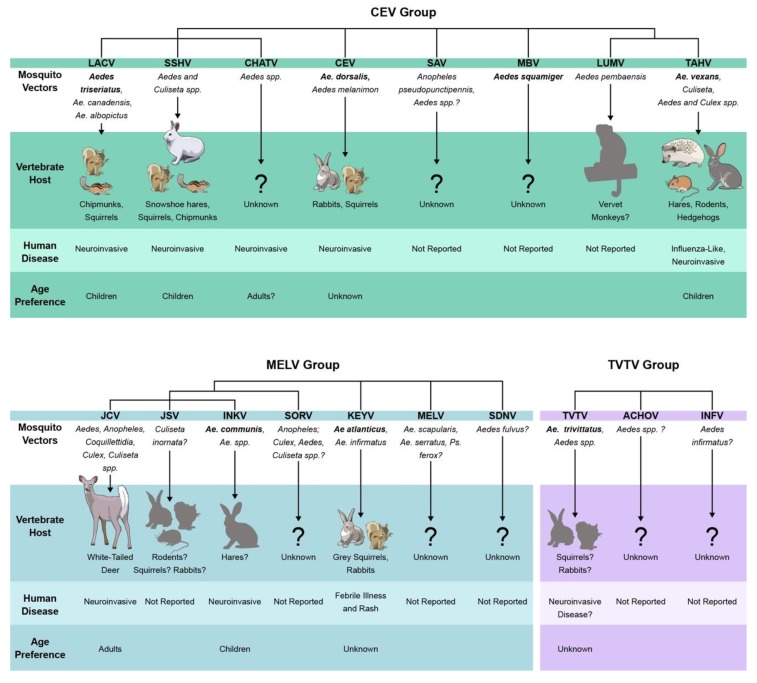
Summary of the genetic relationships, mosquito vectors, vertebrate hosts, and human disease for the California serogroup (CSG) viruses, expanded from Figure 1B in [21]. Groups are divided based on phylogenetic classification. Relationships reflect the compilation and summary of several phylogenetic analyses [8,14,18,19,20]. Monophyletic groups are shown and reflect that these were consistent in the majority of studies, and ungrouped vertical lines indicate relationship discrepancies of those viruses between phylogenetic studies. Mosquito genera and species from which CSG viruses have been isolated from are listed, and species in bold indicate primary vectors as determined by frequent virus isolations and laboratory transmission studies. Confirmed and likely vertebrate hosts are designated with colored icons and are based on evidence of both serological evidence in the field, and experimental viremias or virus isolations from animals in the field. Grayed icons indicate possible hosts based on either serological evidence or experimental viremias.

**Figure 2 viruses-11-00794-f002:**
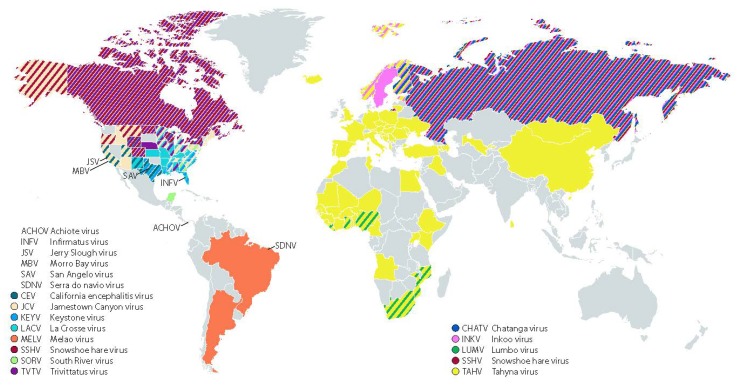
Map of the geographical distribution of the California serogroup viruses, expanded from Figure 1A in [21]. Geographic distribution is based on positive serology of humans and animals, mosquito isolations of virus, and probable and confirmed human cases.

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
