# Peer review of "Throw out the Map: Neuropathogenesis of the Globally Expanding California Serogroup of Orthobunyaviruses"

_viruses, 2019, doi:10.3390/v11090794_

Round 1
Reviewer 1 Report
The orthobunyaviruses in the California serogroup are distributed global and many are reported neuropathogenic to human. The review by Evans and Peterson covers their geological distribution, vector and host range, viral replication, infection in animal and human, and the current knowledge about the mechanisms of neuropathogenesis. The review is well written and provides readers comprehensive information of CSG viruses.
There are a few points to consider for correction and improvement.
Page 1. The authors’ affiliations are identical. Line 145 Page 4. Use full name for Nab (neutralizing antibody) when it first appears in text. Lines 166 to 168 page 5. The viral entire RNA segments are encapsidated by nucleoprotein (N) to form RNP. The interaction between viral genomic RNA and N goes through whole segment rather than only in UTRs. It is supposed that the L protein is associated with the RNA segment at UTR region. Line 168 page 6. I would suggest use ‘L protein’ instead of the RdRp in the text in lines 168 and 169. RdRp is the major function of L protein, but L protein has also endonuclease domain at the its N-terminal domain which is involved in the “cap-snatching’ activity. Line 172 page 6. The cleavage of orthobunyavirus glycoprotein precursor (GPC) into mature products, Gn, Gc and NSm is one of important events in bunyaviral replication. The GPC processing is mediated by cellular signal peptidase and signal peptide peptidase (SPP). Line 195 page 6. “Orthobunyavirus virions assemble …bud from the Golgi.” It would be more accurate for the text to be changed to “at the membrane of the Golgi.” Line 252 page 7. Remove ‘widely’. Line 271 pate 8. Consider change ‘neuropathogenesis’ to ‘neuropathogenicity’.
Author Response
Reviewer 1:
Comments and Suggestions for Authors
The orthobunyaviruses in the California serogroup are distributed global and many are reported neuropathogenic to human. The review by Evans and Peterson covers their geological distribution, vector and host range, viral replication, infection in animal and human, and the current knowledge about the mechanisms of neuropathogenesis. The review is well written and provides readers comprehensive information of CSG viruses.
There are a few points to consider for correction and improvement.
Point 1) Page 1. The authors’ affiliations are identical.
Response 1) We have combined the affiliations and listed both emails after affiliation 1.
Point 2) Line 145 Page 4. Use full name for Nab (neutralizing antibody) when it first appears in text.
Response 2) Thank you for catching this oversight. Neutralizing antibody has now been spelled out in its first use (now line 241).
Point 3) Lines 166 to 168 page 5. The viral entire RNA segments are encapsidated by nucleoprotein (N) to form RNP. The interaction between viral genomic RNA and N goes through whole segment rather than only in UTRs. It is supposed that the L protein is associated with the RNA segment at UTR region.
Response 3) Thank you for catching this mistake, the text has now been updated to read “Encapsidation of the genome segments is initiated by interactions between the 5’ end of the genomic segments and the Nucleocapsid (N) protein, resulting in ribonucleoprotein (RNP) complexes. These genomic RNPs associate with the L protein in the virion through interactions with the UTRs.” Now lines 273-276.
Point 4) Line 168 page 6. I would suggest use ‘L protein’ instead of the RdRp in the text in lines 168 and 169. RdRp is the major function of L protein, but L protein has also endonuclease domain at the its N-terminal domain which is involved in the “cap-snatching’ activity.
Response 4) We have replaced RdRp with “L protein”, now lines 276-277.
Point 5) Line 172 page 6. The cleavage of orthobunyavirus glycoprotein precursor (GPC) into mature products, Gn, Gc and NSm is one of important events in bunyaviral replication. The GPC processing is mediated by cellular signal peptidase and signal peptide peptidase (SPP).
Response 5) We have updated the text to include the important discovery of the M polyprotein cleavage mechanism, and the text now reads “M encodes a single open-reading frame (ORF) resulting in the glycoprotein precursor (GPC) polyprotein in the order Gn-NSm-Gc [117]. In studies of BUNV, it was shown that GPC is processed into mature proteins by internal signal peptides and host signal peptide peptidase [118].” This is now lines 281-282.
Point 6) Line 195 page 6. “Orthobunyavirus virions assemble …bud from the Golgi.” It would be more accurate for the text to be changed to “at the membrane of the Golgi.”
Response 6) This line has been updated to read “Orthobunyavirus virions assemble, mature, and bud at the membrane of the Golgi [106]”. Now line 306.
Point 7) Line 252 page 7. Remove ‘widely’.
Response 7) Removed (now line 382).
Point 8) Line 271 pate 8. Consider change ‘neuropathogenesis’ to ‘neuropathogenicity’.
Response 8) “Neuropathogenesis” was changed to “neuropathogenicity”, now line 401.
Reviewer 2 Report
The review "Throw out the map: neuropathogenesis of the globally expanding California serogroup of orthobunyaviruses" is a useful summary of the state of knowledge of this understudied group of viruses. In general it seems thorough and comprehensive though I do have some suggestions to improve the presentation and aid the reader in what is admittedly a confusing variety of viruses.
Personally I think it would be easier to read if you used the full names of the various viruses throughout rather than the acronyms (it is a very long list of acronyms to wade through otherwise) The virus classification section really should include phylogenetic trees of the different virus segments as it's quite hard to follow otherwise (this can be done relatively easily with at least available reference sequences) . Similarly a table with a comparison of the serogroup and genetic classifications for comparison would be helpful. The information on virus reassortment that is currently in the conclusion needs moving up to this section. It is considered poor writing style to introduce new information in a conclusion and it belongs up here to put your discussion about variations in the classification in context (if your segments reassort incongruence in trees of different virus segments is to be expected). line 61-64, this bit isn't entirely clear - are you talking about differing positions in trees in different papers? (having your own trees would also clarify this) you haven't really gone into replication and immune response in the mosquitoe host at all (aside from the transovarial transmission) is there much known on the mosquitoe side? line 140. What is known about JCV in domestic ruminants? Throughout this entire section you need to be more specific about your vertebrate host species - please give latin and common names. "Rodent" in particular is just not specific enough, do you mean there is experimental evidence that you can infect house mice or is there actual field evidence and in which species? This matters as if the host species is geographically restricted the virus may well be too. It would be helpful to put the geographic range in the tables in figure 2 as well as in figure 1 The replication section could really do with a lifecycle diagram to make it easier to follow. Similarly a diagram of the genetic layout and the virion structure would also be helpful DC-SIGN is a pathogen recognition molecule (which always makes me a bit dubious when it is identified as a specific virus receptor - you would expect it to bind to many pathogens) Any evidence of pathology in non-human hosts (?) aside from mouse models of course. line 208-220, these sentences are very awkward. line 216. Do they just cross react wiht other california serogroup viruses or with other bunyaviruses? (is that known?) also line 216 by "early cases" do you mean historical cases or the intitial phase after viral infection in conteporary cases? line 270 there appears to be a repeated sentence here. line 309. This can be deleted as you've already said that elsewhere I would change the title of the neuropathogenesis section to "Immunopathogenesis" as most of what you are talking about here are immune responses (not pathology descriptions or clinical progression) You could probably delete most of the paragraph 321-331 as this appears to be a basic description of the IFN system and is probably unnecessary here In the next paragraph however you do need to define your acronyms (eg IFNar -/-) line 336 rephrase as follows: dult Irf3/Irf7-/-, Mavs-/- (RLR signalling pathway deficient) and Unc93b13D ( TLR3/7/9 receptor signalling pathway deficient) mice but not Myd88-/- mice (only deficient in TLR signalling) ..... , line 352 probably important to point out that ABV doesn't affect people and LACV doesn't affect ruminants - also define SBV (orthobunyavirus) otherwise this acronym comes out of no-where and its not clear why you would be using this virus as a comparison line 359 is it known in which cell types this occurs? line 378, you probably need to point out what SARM1 does and why this molecule was investigated in this context line 387, similarly define acronym and give basic function of CCR2 line 451. You probably also need to include in your discussion something about why most humans have subclinical disease and only a small number go on to develop encephalitis (age, genetics etc)
Author Response
Reviewer 2:
Comments and Suggestions for Authors
The review "Throw out the map: neuropathogenesis of the globally expanding California serogroup of orthobunyaviruses" is a useful summary of the state of knowledge of this understudied group of viruses. In general it seems thorough and comprehensive though I do have some suggestions to improve the presentation and aid the reader in what is admittedly a confusing variety of viruses.
Point 1) Personally I think it would be easier to read if you used the full names of the various viruses throughout rather than the acronyms (it is a very long list of acronyms to wade through otherwise)
Response 1) While we appreciate that the long list of acronyms can be a little burdensome, we feel that spelling out the full names for all of the viruses throughout the manuscript would bog down the text and make it difficult to read. The virus acronyms are given both in the text as well as in the (now) Figure 2 legend, so they can be easily referenced by the reader.
Point 2) The virus classification section really should include phylogenetic trees of the different virus segments as it's quite hard to follow otherwise (this can be done relatively easily with at least available reference sequences). Similarly a table with a comparison of the serogroup and genetic classifications for comparison would be helpful.
Response 2) Because this is a review, we do not feel it is appropriate to conduct a new phylogenetic analysis, but instead chose to summarize comprehensive phylogenies from recent papers. We did attempt to summarize the relationships in Figure 2, however we admit this was not adequately explained (and is not, of course, a true phylogeny). We do appreciate the reviewer’s comments that this section is hard to follow, and have made several attempts to try to clarify the section in lieu of making a table. The serological and phylogenetic classifications were reordered so that viruses appear in the same order within the text, in an attempt to make it easier to compare group members. The comment that genetic and antigenic relatedness doesn’t necessarily correlate was clarified by explaining only INKV is classified in different groups based on serology (CEV group) and phylogenetic analysis (MELV group). We have also referenced Figure 2 (now Figure 1) for a visual of the virus’s genetic relationships based on compiling and summarizing the findings of the 4 recent phylogenetic studies. This information has also been added to the (now) Figure 1 legend. (Current lines 49-76).
Point 3) The information on virus reassortment that is currently in the conclusion needs moving up to this section. It is considered poor writing style to introduce new information in a conclusion and it belongs up here to put your discussion about variations in the classification in context (if your segments reassort incongruence in trees of different virus segments is to be expected).
Response 3) We disagree with the reviewer that it “is poor writing style” to bring up the information about reassortments in the conclusion. None of the phylogenetic analyses has found strong evidence of reassortment between the 18 main CSG viruses, so we don’t feel this information is relevant in the genetic relationships section. The possibility of reassortment greatly increases with the expanding mosquito and host ranges for these viruses, which makes it very appropriate for the future concerns. However, to address the reviewers concerns, we have retilted this section as “Conclusions and Future Perspectives” so that we are not bringing up new information in a section entitled “Conclusions”.
Point 4) line 61-64, this bit isn't entirely clear - are you talking about differing positions in trees in different papers? (having your own trees would also clarify this)
Response 4) We have updated the text in this portion to clarify that the statements in this section are based on a comparison of the four studies with the most complete CSG virus phylogenies and referred the reader to the updated Figure 1 for a visual summary of these relationships (current lines 68-76, Figure 1).
Point 5) you haven't really gone into replication and immune response in the mosquitoe host at all (aside from the transovarial transmission) is there much known on the mosquitoe side?
Response 5) While there are some studies looking at CSG virus replication in mosquitoes, the vast majority focused on transovarial transmission, which is what we chose to report. We believe that while replication within the mosquito is an important aspect of any mosquito-borne virus, it is beyond the focus of this review which is focused on the neuropathogenesis of these viruses.
Point 6) line 140. What is known about JCV in domestic ruminants?
Response 6) We have updated this line (now line 195) to include the information that NAb to JCV has been detected in domestic animals such as sheep, cattle, and horses.
Point 7) Throughout this entire section you need to be more specific about your vertebrate host species - please give latin and common names. "Rodent" in particular is just not specific enough, do you mean there is experimental evidence that you can infect house mice or is there actual field evidence and in which species? This matters as if the host species is geographically restricted the virus may well be too. It would be helpful to put the geographic range in the tables in figure 2 as well as in figure 1
Response 7) Depending on the virus and particular study, the evidence for suggested hosts is either from serosurveys for neutralizing antibody or viremia in wild animals, or viremia in experimental infections in animals. We have updated the (now) Figure 1 legend to indicate that probable/confirmed hosts are animals that have evidence of both NAb from field serosurveys and experimental viremia or virus isolation from field animals. Suspected hosts (shaded gray) have either field serosurvey evidence or experimental viremia.
We appreciate the reviewer’s comment that “rodent” is not a specific term, however, we believe that listing each specific animal implicated as a potential vertebrate host (of which there are dozens between the 18 CSG viruses) would be very difficult and unpleasant to read. The point of this section is to highlight and generalize the similarities and differences in types of hosts between the CSG viruses, mainly that the majority of CSG viruses utilize small mammals as hosts, except for JCV which uses large mammals. Of course, different species are susceptible to different viruses. However, comprehensive and complete serosurveys and laboratory infections demonstrating viremia have not been conducted for most of the viruses, so the exact restriction of host species is unknown (i.e. simply because viremia has been demonstrated in one squirrel species does not mean another squirrel species is not susceptible). We have summarized this information as concisely as possible as this information alone could be a separate review.
In terms of the host species geographically restricting virus, we have updated line 111 to include vertebrate hosts as well as mosquito vectors. The map (now Fig. 2) includes evidence of virus presence via mosquito collections, serosurveys, and human disease cases. We have tried to be as thorough as possible to indicate geographic restrictions, but of course we are limited by where studies have been conducted and what viruses they were looking for. We have chosen not to add the geographic information to (now) Figure 1 because it would be redundant and clutter the figure.
Point 8) The replication section could really do with a lifecycle diagram to make it easier to follow. Similarly a diagram of the genetic layout and the virion structure would also be helpful
Response 8) Because only a few studies have specifically looked at CSG viruses, and most that do focus only on LACV, what is known and inferred about the replication cycle and much of the virion structure is based on LACV and other bunyaviruses, particularly Bunyamwera virus. There is very little known about any differences in lifecycle between the CSG viruses, so we did not focus on this aspect of the viruses in this review. A very good review of the complete orthobunyavirus replication cycle, genetic structure, and virion structure has already been published (2014 Nature Microbiology Review by Elliott et al.) which we refer readers to.
Point 9) DC-SIGN is a pathogen recognition molecule (which always makes me a bit dubious when it is identified as a specific virus receptor - you would expect it to bind to many pathogens)
Response 9) We agree with the reviewer, however because little is known on the CSG virus (and orthobunyaviruses in general) receptor, we feel the paper that implicated DC-SIGN should be included. We did provide the caveat “However, DC-SIGN is likely not the only, or primary, receptor as its expression is limited to certain Dendritic cell and macrophage types and LACV has a wide range of cell tropism and, in the CNS, primarily infects neurons [121], [122]” (current lines 249-250) to point out to the reader that there are caveats that this is the specific receptor for these viruses.
Point 10) Any evidence of pathology in non-human hosts (?) aside from mouse models of course.
Response 10) To our knowledge no one has published on the pathology in non-human hosts other than the mouse. Experimental studies that have inoculated non-lab mouse animals have not reported disease in these animals, except after direct intracranial inoculation in some cases. Disease in naturally infected animals has not been reported.
Point 11) line 208-220, these sentences are very awkward.
Response 11) We have altered these lines to clarify the meaning of the sentences (current lines start at 290).
Point 12) line 216. Do they just cross react with other california serogroup viruses or with other bunyaviruses? (is that known?)
Response 12) The CSG viruses are classified as such due to their cross reactiivity with each other and little or no cross-reaction with other bunyaviruses. The minor exception to this for the CSG is Guaroa virus which was briefly categorized as a California serogroup virus due to some cross-reactivity, however it is now believed to be a reassortant between a California group and a Bunyamwera group virus. This is described in lines 31-32 of the manuscript.
Point 13) also line 216 by "early cases" do you mean historical cases or the intitial phase after viral infection in conteporary cases?
Response 13) We have changed “early cases” to “historical cases” to clarify this point (current line 299).
Point 14) line 270 there appears to be a repeated sentence here.
Response 14) The section including line 270 (now line 358): “Neuroinvasion describes the ability of the virus to enter the brain from the periphery, regardless of the development of disease. Neuroinvasive disease is evaluated by the ability of the virus to enter the CNS from the periphery and cause disease. Neurovirulence describes the ability of the virus to replicate in the brain following a direct route of inoculation, such as IC or intranasal (IN), and neuropathogenesis is the ability of the virus to cause neurological disease.”
This section describes four different terms: 1) Neuroinvasion; 2) Neuroinvasive disease; 3) Neurovirulence; and 4) Neuropathogenesis. These are related, but distinct, terms which we felt needed to be defined so that our subsequent usage was clear.
Point 15) line 309. This can be deleted as you've already said that elsewhere
Response 15) We prefer to leave this in, because the earlier statement referred to historical cases, and the statement in (current) lines 401-402 discusses the ongoing CSG virus diagnostic challenges.
Point 16) I would change the title of the neuropathogenesis section to "Immunopathogenesis" as most of what you are talking about here are immune responses (not pathology descriptions or clinical progression)
Response 16) Neuropathogenesis is more appropriate, as we discuss both the immune response, the neuronal response and the viral factors that are involved in infection and CNS disease. Immunopathogenesis, therefore, would not be appropriate. Neuropathogenesis is not limited to pathology or clinical description and can include mechanisms of how neurons are damaged during disease.
Point 17) You could probably delete most of the paragraph 321-331 as this appears to be a basic description of the IFN system and is probably unnecessary here In the next paragraph however you do need to define your acronyms (eg IFNar -/-)
Response 17) Because this is a general review of the CSG viruses, many readers likely will not be familiar with innate signaling pathways, so we believe that keeping the brief introduction of the interferon system will aid the reader in understanding the results we go on to discuss.
The acronyms have already been defined, therefore we do not need to redefine them.
Point 18) line 336 rephrase as follows: dult Irf3/Irf7-/-, Mavs-/- (RLR signalling pathway deficient) and Unc93b13D ( TLR3/7/9 receptor signalling pathway deficient) mice but not Myd88-/- mice (only deficient in TLR signalling) ..... ,
Response 18) By keeping the interferon summary paragraph, these clarifications are unnecessary as they have already been described.
Point 19) line 352 probably important to point out that ABV doesn't affect people and LACV doesn't affect ruminants - also define SBV (orthobunyavirus) otherwise this acronym comes out of no-where and its not clear why you would be using this virus as a comparison
Response 19) We have clarified the host restrictions of these viruses by adding the statement “In studies comparing restriction of LACV, which infects humans but not ruminants, and SBV, which infects ruminants but not humans,...”. SBV was previously defined in (current) line 252.
Point 20) line 359 is it known in which cell types this occurs?
Response 20) No, but as we explain in the next line (current 452), infectious virus recovered from tissues is likely from blood, as viral antigen staining has not been described in any peripheral tissues.
Point 21) line 378, you probably need to point out what SARM1 does and why this molecule was investigated in this context line 387, similarly define acronym and give basic function of CCR2 line 451.
Response 21) We have added descriptions of SARM1 and CCR2 in current lines 471-472 and 479-485.
Point 22) You probably also need to include in your discussion something about why most humans have subclinical disease and only a small number go on to develop encephalitis (age, genetics etc)
Response 22) We have added this to line 544-545 of the discussion